# Acridone Derivatives from *Atalantia monophyla* Inhibited Cancer Cell Proliferation through ERK Pathway

**DOI:** 10.3390/molecules27123865

**Published:** 2022-06-16

**Authors:** Wen-Yong Gao, Chantana Boonyarat, Pitchayakarn Takomthong, Kusawadee Plekratoke, Yoshihiro Hayakawa, Chavi Yenjai, Rawiwun Kaewamatawong, Suchada Chaiwiwatrakul, Pornthip Waiwut

**Affiliations:** 1Faculty of Pharmaceutical Sciences, Ubon Ratchathani University, Ubon Ratchathani 34190, Thailand; gaowenyong@biocoso.com (W.-Y.G.); rawiwun.k@ubu.ac.th (R.K.); 2Faculty of Pharmaceutical Sciences, Khon Kaen University, Khon Kaen 40002, Thailand; chaboo@kku.ac.th (C.B.); ppitcha.t@gmail.com (P.T.); kusawadee2535@gmail.com (K.P.); 3Institute of Natural Medicine, University of Toyama, Toyama 930-0194, Japan; haya@inm.u-toyama.ac.jp; 4Natural Products Research Unit, Department of Chemistry, Center of Excellence for Innovation in Chemistry, Faculty of Science, Khon Kaen University, Khon Kaen 40002, Thailand; chayen@kku.ac.th; 5Department of English, Faculty of Humanities and Social Sciences, Ubon Ratchathani Rajabhat University, Ubon Ratchathani 34000, Thailand; suchadachai65@gmail.com

**Keywords:** buxifoliadine E, cancer cells, Erk, Akt, p38, apoptosis

## Abstract

The present study aimed to investigate the effect of acridone alkaloids on cancer cell lines and elucidate the underlying molecular mechanisms. The ten acridone alkaloids from *Atalantia monophyla* were screened for cytotoxicity against LNCaP cell lines by a WST-8 assay. Then, the most potential acridone, buxifoliadine E, was evaluated on four types of cancer cells, namely prostate cancer (LNCaP), neuroblastoma (SH SY5Y), hepatoblastoma (HepG2), and colorectal cancer (HT29). The results showed that buxifoliadine E was able to significantly inhibit the proliferation of all four types of cancer cells, having the most potent cytotoxicity against the HepG2 cell line. Western blotting analysis was performed to assess the expression of signaling proteins in the cancer cells. In HepG2 cells, buxifoliadine E induced changes in the levels of Bid as well as cleaved caspase-3 and Bax through MAPKs, including Erk and p38. Moreover, the binding interaction between buxifoliadine E and Erk was investigated by using the Autodock 4.2.6 and Discovery Studio programs. The result showed that buxifoliadine E bound at the ATP-binding site, located at the interface between the N- and C-terminal lobes of Erk2. The results of this study indicate that buxifoliadine E suppressed cancer cell proliferation by inhibiting the Erk pathway.

## 1. Introduction

According to the WHO’s statistics on the global cancer epidemic, cancer is the second-leading cause of death globally and was responsible for an estimated 10.0 million deaths in 2020. Approximately 70% of the deaths from cancer occur in low- and middle-income countries. The most common causes of cancer death are cancers of the lung (18%), colorectal region (9.4%), liver (8.3%), stomach (7.7%), and female breast (6.9%) [1].

Around one-third of deaths from cancer are due to the five leading behavioral and dietary risks: a high body mass index, low fruit and vegetable intake, lack of physical activity, tobacco use, and alcohol use. Tobacco use is the most important risk factor for cancer and is responsible for approximately 22% of cancer deaths [2,3,4,5,6,7].

The traditional methods for cancer treatment provide unfavorable outcomes for most patients due to their serious side effects [8,9,10]. Therefore, many studies have attempted to discover new therapeutic agents for the management of cancer.

Nowadays, natural products from plants are important as a promising research area for cancer prevention and treatment. Acridone alkaloids are naturally occurring phytochemicals containing the acridin-9(10H)-one. Acridone is a unique alkaloid known to possess several biological functions, such as anti-viral [11], anti-microbial [12], anti-pruritic [13], and anti-Alzheimer’s disease [14,15] activities. Moreover, some acridone derivatives have been shown ability to combat cancer cells [16,17]. Hence, acridone alkaloids might be potential for cancer treatment. In a recent study, we isolated 11 acridone alkaloids and five limonoids from the stems of *Atalantia monophyla*. From preliminary screening, we found that acridone alkaloids showed better cytotoxicity than limonoids [18]. Thus, in this study we focus on studying the effect of acridone alkaloids on cancer cell lines and elucidating the underlying molecular mechanisms.

There are three major pathways for mitogen-activated protein kinases (MAPKs), namely, the p44/42 MAPKs, which are extracellular signal-related kinases (ERKs); the p38 MAPKs; and the stress-activated protein kinases/Jun-N-terminal kinases (SAPKs/JNKs) [19]. They play important roles in cellular programs, such as proliferation, differentiation, development, transformation, and apoptosis. JNK and p38 have been linked to cell death and tumor suppression, whereas ERK plays a role in cell survival and tumor promotion [20]. Moreover, Akt also plays roles in cell metabolism, proliferation, and survival. Both signaling pathways, MAPKs and Akt, are mediated through the serine and/or threonine phosphorylation of a range of downstream substrates, and they have been implicated in the responses of tumor cells as therapeutic targets, suggesting that they may be a promising target for anti-cancer agents [21,22].

Thus, in this study, we aimed to study the effect of acridone alkaloids from *Atalantia monophyla* on cancer cell lines and elucidate the underlying molecular mechanisms. The ten acridone derivatives from *Atalantia monophyla*, namely *N*-methylatalaphylline, atalaphylline, *N*-methylatalaphyllinine, atalaphyllinine, *N*-methylcycloatalaphylline A, citrusinine II, citrusinine I, glycosparvarine, citruscridone, and buxifoliadine E (Figure 1), were screened for antiproliferation activity by WST-8 assay. The most potential acridone was evaluated on four types of cancer cells, namely prostate cancer (LNCaP), neuroblastoma (SH SY5Y), hepatoblastoma (HepG2), and colorectal cancer (HT29). To elucidate the underlying molecular mechanism, the effects of the most potential acridone on expression of proteins involved in promoting proliferation and regulating the *cell cycle* and apoptosis were investigated by Western blotting analysis. In addition, the effect of the acridone on ERK activity was evaluated in vitro and the binding interaction between the acridone and ERK was performed in silico by program AutoDock.

## 2. Results

### 2.1. Cytotoxicity Effect of 10 Acridone Alkaloids

All 10 acridone alkaloids at the concentration of 100 µM were evaluated for anti-proliferation activity by using LNCaP prostate-cancer cell lines. Figure 2 indicated that all acridone alkaloids exhibited the moderate to good cytotoxicity activity, except atalaphyllinine (A4) and citruscridone (A9), which demonstrated no effect on this cell line. Furthermore, buxifoliadine E (A10) was the most potent compound.

### 2.2. Dose-Response and Time-Course Studies

In this study, the most potent acridone, buxifoliadine E was evaluated for anti-proliferation activity of prostate cancer (LNCaP) at various concentrations (0.1 μM, 1 μM, 10 μM, 100 μM) and times (24 h, 48 h, 72 h). Figure 3 indicated that buxifoliadine E significantly reduced LNCaP cell proliferation in a dose- and time-dependent manner. Moreover, the 50% inhibitory concentration range of buxifoliadine E did not change with the treatment duration (24–72 h); the results showed that it ranged from 1 μM to 10 µM. Therefore, the time treatment of the test compound at 24 h was chosen to study in further investigations.

### 2.3. Cytotoxic Effects of Buxifoliadine E on Prostate Cancer (LNCaP), Neuroblastoma (SH SY5Y), Hepatoblastoma (HepG2), and Colorectal Cancer (HT29)

The most potent acridone, buxifoliadine E, was evaluated at various concentrations for anti-proliferation activity of prostate cancer (LNCaP), neuroblastoma (SH SY5Y), hepatoblastoma (HepG2), and colorectal cancer (HT29). After treatment for 24 h, buxifoliadine E showed an ability to enhance cytotoxicity of all LNCaP, HepG2, HT29, and SHSY5Y cancer cells. Comparing among cancer-cell types, buxifoliadine E at 100 µM significantly induced LNCaP, HepG2, HT29, and SHSY5Y cancer-cell death with IC_50_ of 43.10 µM, 41.36 µM, 64.60 µM, and 96.27 µM. respectively, having the most potent cytotoxicity against the HepG2 cell line. (Figure 4, Figure 5, Figure 6 and Figure 7).

### 2.4. Effects of Buxifoliadine E on Apoptotic Signaling Proteins in HepG2 Cells

Since buxifoliadine E at 100 µM demonstrated the most potent effect on HepG2 cells, we decided to investigate the molecular mechanism by which buxifoliadine E induced the death of HepG2 cells. The cells were treated with buxifoliadine E at various concentrations, using 10 μM doxorubicin and the Erk signaling pathway *inhibitor* 10 μM PD98059 (*PD*) as positive controls. The results are represented in Figure 8.

### 2.5. Effects of Buxifoliadine E on the Expression Levels of Proteins Related to ERK/MAPK Signaling Pathway in the HepG2 Cells

*The ERK*/*MAPK* signaling pathway is involved in promoting proliferation and regulating the *cell cycle* and apoptosis. p38 MAPK can play a role as a regulator of cell death; thus, it can mediate cell death through different mechanisms, including apoptosis. We investigated the effects of buxifoliadine E on the phosphorylation of Erk and p38 proteins, using 10 μM doxorubicin and 10 μM PD as positive controls. Figure 9 shows that buxifoliadine E at a concentration of 100 μM markedly inhibited the phosphorylation of Erk, which correlated with Erk inhibition by PD, but doxorubicin did not affect Erk phosphorylation. Regarding p38, buxifoliadine E increased the phosphorylation of p38 in a concentration-dependent manner; doxorubicin and PD, which were the positive controls, also activated the phosphorylation of p38. This indicates that buxifoliadine E induces apoptosis through the Erk/MAPK pathway.

### 2.6. Buxifoliadine E Inhibited Erk Kinase Activity

The effect of buxifoliadine E on Erk kinase-enzyme activity was performed by using an Erk kinase-enzyme system with an ADP-Glo^TM^ kinase assay. Bufoliadine E at concentrations 1 μM, 10 μM, and 10 μM significantly inhibited Erk kinase-enzyme activity, when comparing with an Erk inhibitor (PD98059) (Figure 10). The result indicated that buxifoliadine E works as an Erk kinase inhibitor.

### 2.7. Binding Interaction Studies between Buxifoliadine E and Target Erk

To further clarify mechanism of action, the binding interaction between buxifoliadine E and Erk was investigated by using the Autodock 4.2.6 and Discovery Studio programs. The binding modes and interaction diagrams of buxifoliadine E and PD bound to Erk2 are shown in Figure 11 and Figure 12, respectively.

### 2.8. Pharmacokinetic and Toxicity Profiles of Buxifoliadine E

The pharmacokinetics properties (absorption, distribution, metabolism, excretion, ADMET) and toxicity profile of buxifoliadine E were studied by the in silico method. The parameters for evaluating ADMET were estimated using pKCSM and are shown in Table 1. The in silico investigation of ADMET revealed that human-intestinal absorption of buxifoliadine E is 68.876%. The steady-state volume of distribution of buxifoliadine E was 1.8 L/kg. Buxifoliadine E showed no effect on almost all CYP types except CYP2C19. The excretion and toxicity of buxifoliadine E were also analyzed.

## 3. Discussion

The acridone alkaloids exhibit numerous biological functions, such as anti-viral [23] anti-malarial, anti-allergy [24], and anti-cancer activities. However, in cancer, the inhibitory effects of acridone alkaloids are not fully understood. Therefore, the aim of the present study was to determine the effects of acridone alkaloids on cancer cells and to elucidate the underlying molecular mechanisms.

All 10 acridone alkaloids from *Atalantia monophyla*, namely *N*-methylatalaphylline, atalaphylline, *N*-methylatalaphyllinine, atalaphyllinine, *N*-methylcycloatalaphylline A, citrusinine II, citrusinine I, glycosparvarine, citruscridone, and buxifoliadine E, were screened for cytotoxicity against LNCaP cell lines using WST-8 assay. Almost all compounds at the concentration of 100 μM exhibited moderate to good cytotoxicity activity, except atalaphyllinine and citrucridone, which demonstrated no effect on this cell line. Among all acridones, buxifoliadine E exhibited the most potent activity with a percentage of cell viability of 20.88. The structure–cytotoxic activity relationship study reviewed that the replacement of hydroxy group with methoxy group at position 3 of acridone ring seemed to diminish cytotoxic activity. Apparently, citrusinine I (cell viability = 69.32%) possessed less activity than citrusinine II (cell viability = 54.09%). In addition, the presence of methoxy group at position 2 of citrucidine (cell viability = 90.56%) resulted in reduced activity, comparable to citrucinine II. Substitution of methoxy group at position 4 with hydroxy group enhanced the cytotoxicity, and apparently glycosparvaline (cell viability = 50.94%) provided higher activity than citrucridone. Regarding substituents at the position 10 of the acridone ring, the substitution of a methyl group resulted in an activity increase. N-methylatalaphylline (cell viability = 33.07%) and N-methylatalaphyllinine (cell viability = 89.46%) were slightly more potent than atalaphylline (cell viability = 48.32%) and atalaphyllinine (cell viability = 97.37%), respectively. Cyclization at positions 3–4 on the acridone alkaloid ring proved unfavorable to cytotoxicity activity. Atalaphyllinine and N-methylatalaphyllinine showed weaker activity than atalaphylline and N-methylatalaphylline, respectively. However, the cyclization at position 2–3 did not affect cytotoxic activity. Apparently, N-methylatalaphylline and N-methylcycloatalaphylline A possessed similar activities. Interestingly, the presence of a furan ring at positions 2–3 on the acridone ring showed an enhanced cytotoxic activity, as shown by the increased activity of buxifoliadine E.

Among the 10 acridones, buxifoliadine E showed the most potent cytotoxicity and significantly reduced LNCaP cell proliferation in a dose- and time-dependent manner. Thus, buxifoliadine E was chosen to determine antiproliferation against various cancer cell types and to elucidate the underlying molecular mechanisms. Previous study showed that buxifoliadine E exhibited cytotoxicity against KKUM156 and HepG2 cell lines, with IC_50_ values ranging from 2 µg/mL to 3.8 µg/mL [18,25,26]. However, the mechanism of cytotoxicity has not been elucidated. Thus, in the present study, buxifoliadine E was investigated for antiproliferation against various cancer cell types, and its underlying molecular mechanism was elucidated.

For anti-proliferation activity against various cancer cells, after treatment for 24 h, buxifoliadine E showed an ability to enhance cytotoxicity of LNCaP, HepG2, HT29, and SHSY5Y cancer cells. Comparing among cancer-cell types, buxifoliadine E at the concentration of 100 µM showed the most potent cytotoxicity against the HepG2 cell line.

To identify the cytotoxicity mechanisms of buxifoliadine E, we investigated the effect of buxifoliadine E on the expression of pro- and anti-apoptotic proteins using Western blot analysis. The results showed that buxifoliadine E induced increased levels of Bax, a proapoptotic protein, and cleaved caspase-3, a marker of apoptosis, while it reduced the level of Bid in a concentration-dependent manner. These results indicate that buxifoliadine E induces cancer-cell death through an apoptotic pathway.

Thus, we further investigated the molecular mechanisms of buxifoliadine E, and the results revealed that buxifoliadine E affected the function of ERK proteins and seemed to induce p38α protein expression (Figure 9).

The MAPK and Akt pathways frequently malfunction in cancer; they are involved in the upregulation of Akt and MAPKs/ERKs and lead to the promotion of the survival of cancer cells. In the case of the MAPK/p38 pathway, these effects are also related to the downregulation of p38 and suppression of the apoptotic process, which consequently promotes tumor aggressiveness [22].

Extracellular regulated kinase (ERK) is an important kinase in the pathway downstream of the Ras-Raf-MEK-ERK signal transduction pathway. Erk functions as a central link of multiple signaling pathway. The activation of Erk phosphorylates serine/threonine residues of more than 50 downstream cytosolic and nuclear substrates, leading to alteration in gene expression profiles and an increase in proliferation, differentiation, and cell survival. Thus, ERK is an interesting target for the development of potential anti-cancer therapeutic. Erk inhibitors may be promising as an effective cancer treatment that can against cancers with Erk pathway. BVD-523 and GDC0994 are two ERK inhibitors, which have reached clinical trial. BVD-523 has strong pharmacodynamic effects against *p*-Erk and downstream substrates. GDC-0994 is a very potent dual inhibitor of ERK1/2, now in the recruiting stage of a dose-escalation trial in patients with solid tumors. SCH-772984 showed the ability to inhibit phosphorylation of the activation loop of Erk by MEK in A375 melanoma cells [27]. PD98059, as a specific small molecular inhibitor of ERK1/2, inhibited ERK1/2 to block the ERK pathway. It has been shown that PD98059 and the effect of PD98059 in combination with paclitaxel inhibited cancer cells of CRC patients [28].

Our in vitro result showed that the buxifoliadine E inhibited Erk. To clarify the mechanism of action, the binding interaction between buxifoliadine E and Erk was studied in silico by using a molecular docking. The docking result indicated that both the Erk inhibitor PD and buxifoliadine E bound at the ATP-binding site were located at the interface between the N- and C-terminal lobes of Erk2, establishing several hydrogen bonds and hydrophobic interactions with gatekeeper residue, hinge region, salt-bridge region in catalytic segment, KDD motif of the catalytic segment, and other nearby residues. Buxifoliadine E interacts with Erk2 at ATP-binding site by several hydrogen-bonding and hydrophobic interactions. Buxifoliadine E consists of an acridone ring fused with a furan ring at positions 2 and 3, resulting in a planar structure. The binding-orientation analysis revealed that the planar structure of the acridone ring allows the establishment of a π–π stacking interaction with Asp106, Leu107, and Met108 in the hinge region. The furan ring of buxifoliadine E was incorporated into the salt-bridge region in catalytic segment by forming hydrogen bond with Lys54 and Glu71, interacting with Asp167 in the KDD motif of the catalytic segment. Lys54 is known to be a key residue for ATP binding, by helping position the ATP for catalysis. The presence of a salt-bridge between Lys54 and Glu71 lead to the formation of the activated state of Erk. The KKD motif, which is composed of Lys54, Asp167, and Asp149, plays a role in the catalytic properties of Erk. Thus, interrupting salt-bridge interaction and blocking the catalytic process might lead to the inhibition of Erk function. Moreover, the hydroxy group at position 2 and NH at position 10 of the acridone ring formed hydrogen bonds with Gln105, a gatekeeper residue that plays a role in determining the selectivity of kinase inhibitors. Furthermore, buxifoliadine E also interact with other residues: Ile103, Ser153, Leu156, and Cys166. Thus, the docking results confirmed that buxifoliadine E was located at the ATP-binding site, thereby interfering Erk function. The result in Figure 10 confirms that buxifoliadine E reduced Erk kinase activity.

Furthermore, the pharmacokinetics properties (absorption, distribution, metabolism, excretion, ADMET) and toxicity profile of buxifoliadine E were studied. The ADMET from the in silico investigation showed that human-intestinal absorption of acridones ranged from 68.876% to 99.555% (15). In this study, the parameters for evaluating ADMET were estimated using pKCSM web server [29]. The in silico investigation of ADMET revealed that human intestinal absorption of buxifoliadine E is 68.876%. The distribution of buxifoliadine E was considered by the volume of distribution (VDss). The VDss values of buxifoliadine E was 1.8 L/kg. The metabolism and excretion of the acridones were also analyzed. Cytochrome P450 is a superfamily of important detoxification enzymes in the body that are primarily located in the liver and the small intestine. They oxidize xenobiotics and play a role in their clearance. The inhibition of the CYP isoforms of cytochrome P450 by buxifoliadine E is shown in Table 1. Buxifoliadine E showed no effect on all isoforms of cytochrome P450, except CYP2C19. Thus, buxifoliadine E might not affect CYP450 enzyme activity, which suggests that it will not affect the metabolism of the various drug substrates of CYP1A2, CYP2C9, CYP2C6, and CYP3A4. Furthermore, the kidney also plays an important role in drug elimination. The organic cation transporter 2 (OCT2) is a primary renal-uptake transporter that plays a role in the disposition and clearance of organic cation drugs. Buxifoliadine E is not likely to be an OCT2 substrate, as it did not display drug–drug interactions to decrease the renal clearance of an OCT2 substrate. The renal total clearance was also calculated, and the log of the total clearance value of buxifoliadine E is 0.198 mL/min/kg. Moreover, buxifoliadine E showed no hepatotoxicity. In summary, buxifoliadine E showed good ADMET profiles. However, this needs to be confirmed in in vitro and in vivo studies.

## 4. Materials and Methods

### 4.1. Materials

#### 4.1.1. Acridone Alkaloids

The ten acridone alkaloids (Figure 1) extracted from the stem of *Atalantia monophyla*, including N-methylatalaphylline (A1), atalaphylline (A2), N-methylatalaphyllinine (A3), atalaphyllinine (A4), N-methylcycloatalaphylline A (A5), citrusinine II (A6), citrusinine I (A7), glycosparvarine (A8), citruscridone (A9), and buxifoliadine E (A10), were provided by Chavi Yenjai (Faculty of Science, Khon Kaen University, Thailand). The isolation and structural elucidation are described elsewhere [18]. Briefly, *Atalantia monophyla* was collected from Khon Kaen Province, Thailand, and was identified by Dr. Pranom Chantaranothai, Faculty of Science, Khon Kaen University, Thailand. The relative herbarium voucher specimen was deposited at the Faculty of Science, Khon Kaen University, Thailand, with a voucher number of KKU022015. The procedures for isolation of these acridone alkaloids were explained in a previous report [18].

#### 4.1.2. Agents

All the chemicals used in this study were obtained from Invitrogen, Carlsbad, USA. The β-actin, AKT, and p38α were purchased from Santa Cruz Biotechnology, Inc. (Santa Cruz, CA, USA). The *p*-AKT, *p*-p38α, and *p*-ERK were purchased from Cell Signaling Technology, Inc. (Danvers, MA, USA). Rabbit polyclonal antibodies against p38α, *p*-p38α, *p*-Akt, and *p*-ERK were purchased from DAKO (Glostrup, Denmark). Goat polyclonal antibodies against Akt and β-actin were purchased from DAKO.

### 4.2. Cell Culture and Treatment

Human prostate adenocarcinoma (LNCaP prostate), human neuroblastoma (SH SY5Y), human hepatocellular carcinoma (HepG2), and human colorectal adenocarcinoma (HT29) cells were cultured in RPMI-1640 medium (Invitrogen, Carlsbad, CA, USA) with 10% fetal bovine serum (FBS; ICN, Biomedicals, Inc., Aurora, OH, USA), 1 mM L-glutamine (Invitrogen), 100 U/mL penicillin, and 100 µg/mL streptomycin in a humidified atmosphere of 95% air and 5% CO_2_ at 37 °C.

### 4.3. Cell Cytotoxicity

Cells were seeded in 96-well plates at a density of (2.5–5) × 10^3^ cells/well and incubated for 24 h before treatment. The test compounds were prepared at 10 mM as stock solutions. Cancer cells were treated with various concentrations of test compounds for the specified times. The cell viability was determined using WST-8, and the absorbance of the plates at 450 nm was read using a microplate reader. All determinations were carried out at least three times and in triplicate wells.

### 4.4. Western Blot Analysis

Cell lysates were suspended in dilution buffer (20 mM HEPES, pH 7.7, 2.5 mM MgCl_2_, 0.1 mM EDTA, 0.05% Triton X-100, 20 mM β-glycerophosphate, 1 mM sodium orthovanadate, 1 mM PMSF, 1 mM DTT, 10 μg/mL aprotinin, 10 μg/mL leupeptin) on ice. Collected cells were fractured by sonication on ice and then centrifuged at 10,000× *g*, 4 °C, for 15 min. Then, the protein was extracted and denatured at 95 °C for 10 min. The proteins were separated by 10% SDS polyacrylamide gel electrophoresis and then electrophoretically transferred to a nitrocellulose membrane. Subsequently, the membranes were incubated in the presence of different primary antibodies at 4 °C overnight, and then the membranes were incubated with different secondary antibodies for 1 h. Finally, enhanced chemiluminescence (ECL) solution was used for antibody binding and the chemiluminescence of the membranes. The experiment was done in independent triplicates.

### 4.5. Erk Kinase Activity Assay

Erk kinase activity was assessed by using an Erk kinase-enzyme system with an ADP-Glo^TM^ kinase assay (Promega, Madison, WI 53711, USA). In 96-well white plate, perform Erk kinase reaction using reaction buffer A with 50 μM DTT, kinase enzyme, 0.5 μg MBP protein substrate, 50 μM ATP, and inhibitor (DMSO, 1, 10, 100 μM buxifoliadine E and 10 μM PD98059). The process was: incubate at room temperature for 30 min, add 5 μL ADP-Glo^TM^ reagent, incubate for 40 min, add 10 μL kinase detection reagent, and incubate for 30 min. Then, determine luminescence (integration time 0.5–1 s).

### 4.6. In Silico Binding Interaction Studies between Buxifoliadine E and Target Erk

To elucidate the binding interaction via molecular docking studies, an Erk2 template was constructed from an X-ray crystal structure (PDB code: 1TVO) [30]. The constructed Erk2 template was validated by re-docking with a crystallographic structure of Erk2 inhibitor (5-(2-phenylpyrazolo [1,5-a]pyridine-3-yl)-1H-pyrazoro [3,4-C]pyridazin-3-amine, FRZ). All water molecules in the crystal structure of Erk2 were removed, and hydrogen and Gasteiger charges were added by using AutoDockTools (ADT). The AutoGrid was used to generate the grid box with the grid spacing of 0.375 Å, a grid box size of 60 × 60 × 60 Å. Buxifoliadine E and PD (Erk inhibitor) were drawn, and energy was minimized with ChemDraw and Chem3D 15.1 software. A Lamarckian genetic-algorithm protocol was set by using a population size of 100 individuals with 100 ligand-orientation runs. Additionally, the energy evaluation was 1,000,000, and the maximum number of evaluations was 27,000. All ligands were docked by using the Lamarckian genetic algorithm via the Autodock 4.2.6 program. The orientation with the lowest docked energy was considered as the best conformation. After the docking process, the docking complex poses were analyzed for their interactions by using BIOVIA Discovery Studio 2017.

### 4.7. In Silico Pharmacokinetic Properties of Buxifoliadine E

The pharmacokinetic properties (absorption, distribution, metabolism, excretion) and toxicity profile of buxifoliadine E were evaluated by using pKCSM web server [29]. The absorption ability and distribution of buxifoliadine E were predicted based on human-intestinal absorption and the steady state volume of distribution, respectively. The CYP inhibition model including CYP1A2, CYP2C19, CYP2C9, CYP2C6, and CYP3A4 were used to predict metabolism. Excretion was predicted based on the total clearance and renal OCT2 substrate. The toxicity of drugs is predicted based on the hepatotoxicity.

### 4.8. Statistical Analysis

The data were analyzed using IBM *SPSS* statistics Version 24. The statistical technique used for the analysis was a one-way analysis of variance (ANOVA). The *analysis* was performed in *triplicate*, and the values are expressed as the means ± SDs. *p*-values less than 0.05 were considered statistically significant.

## 5. Conclusions

Buxifoliadine E, an acridone alkaloid from *Atalantia monophyla* that acts as Erk inhibitor, inhibited Erk enzyme activity, which causes inhibition of Mcl-1 expression, increasing Bax, which induced caspase-3 activation and apoptosis of cancer cell (Figure 13). Thus, these results suggested that buxifoliadine E is a potential candidate for the development of treatments for cancer. However, further study is still needed to provide more data for the investigation of the specific signaling pathway.

## Figures and Tables

**Figure 1 molecules-27-03865-f001:**
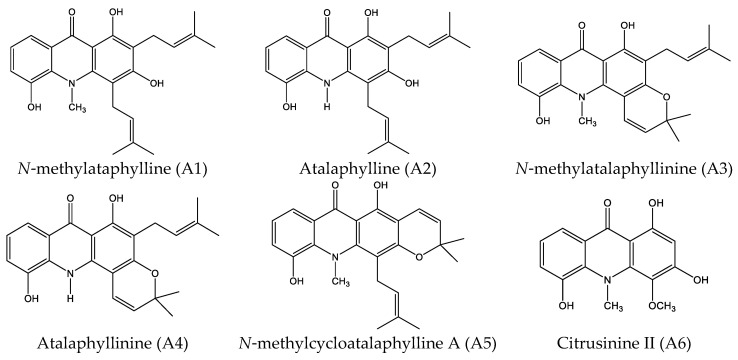
Structures of acridone alkaloids.

**Figure 2 molecules-27-03865-f002:**
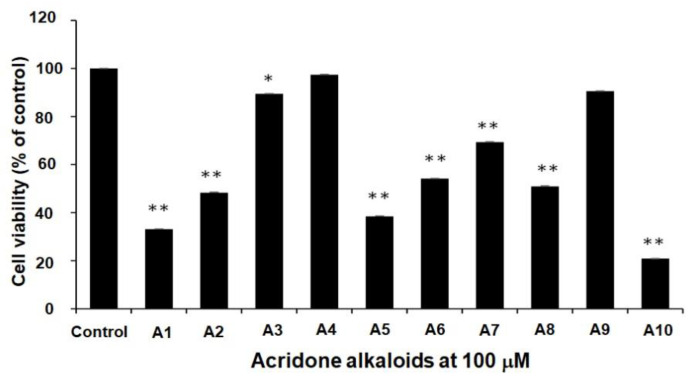
The cytotoxicity effect of acridone alkaloids N-methylatalaphylline (A1), atalaphylline (A2), N-methylatalaphyllinine (A3), atalaphyllinine (A4), N-methylcycloatalaphylline A (A5), citrusinine II (A6), citrusinine I (A7), glycosparvarine (A8), citruscridone (A9), and buxifoliadine E (A10)) on LNCaP cell line. Data are the means ± SD of at least three independent experiments; * *p* < 0.05, ** *p* < 0.01 by One-way ANOVA.

**Figure 3 molecules-27-03865-f003:**
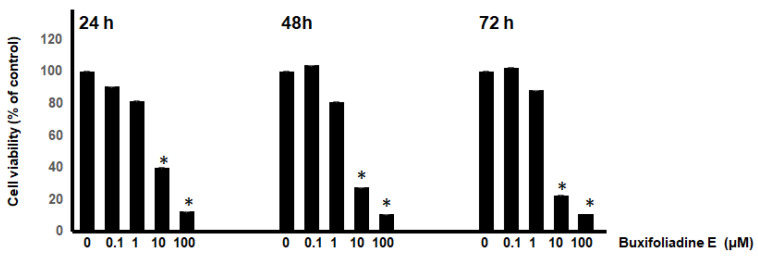
Dose-response and time-course experiment. LNCaP cells were treated with various concentrations of buxifoliadine E for 24 h, 48 h, and 72 h. * *p* < 0.05.

**Figure 4 molecules-27-03865-f004:**
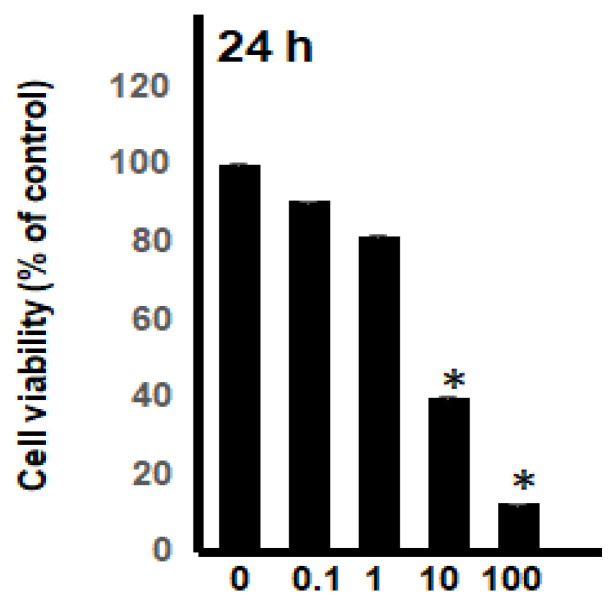
Effect of buxifoliadine E on LNCaP cancer cell death. * *p* < 0.05.

**Figure 5 molecules-27-03865-f005:**
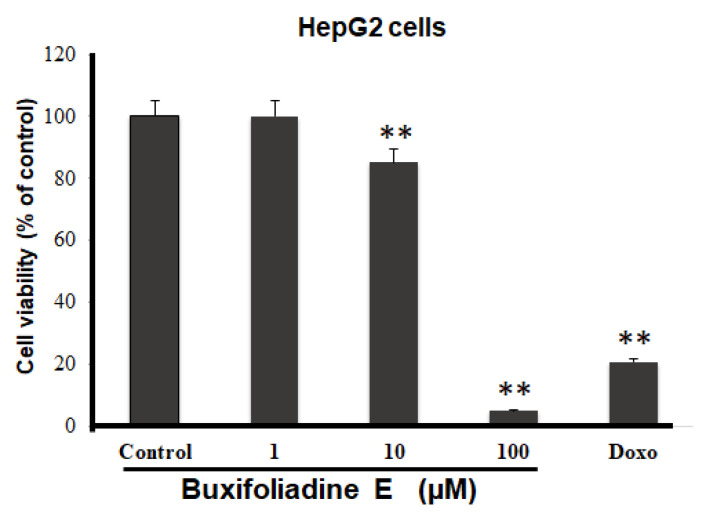
Effect of buxifoliadine E on HepG2 cancer cell death. ** *p* < 0.01.

**Figure 6 molecules-27-03865-f006:**
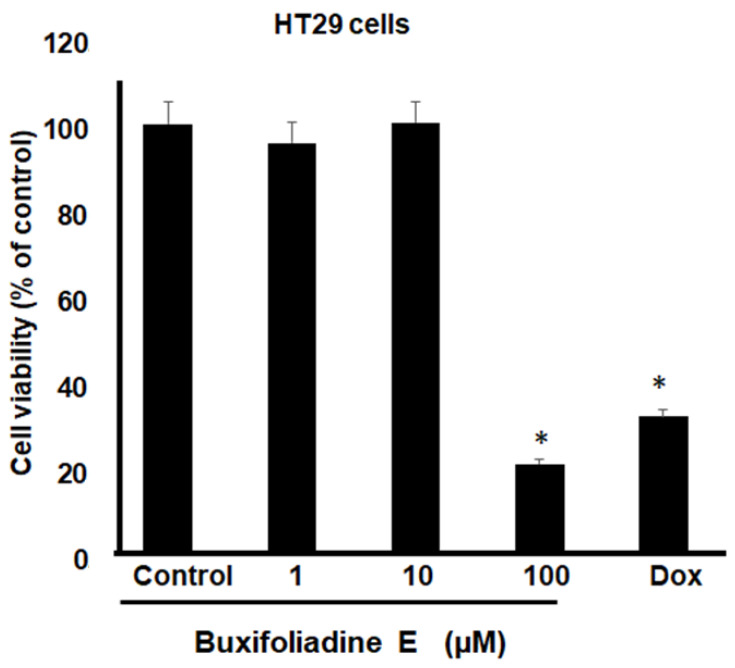
Effect of buxifoliadine E on HT29 cancer cell death. * *p* < 0.05.

**Figure 7 molecules-27-03865-f007:**
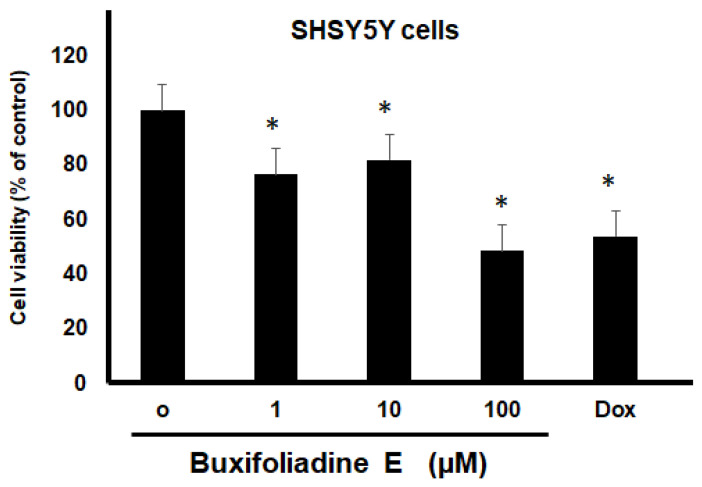
Effect of buxifoliadine E on SHSY5Y cancer cell death. * *p* < 0.05.

**Figure 8 molecules-27-03865-f008:**
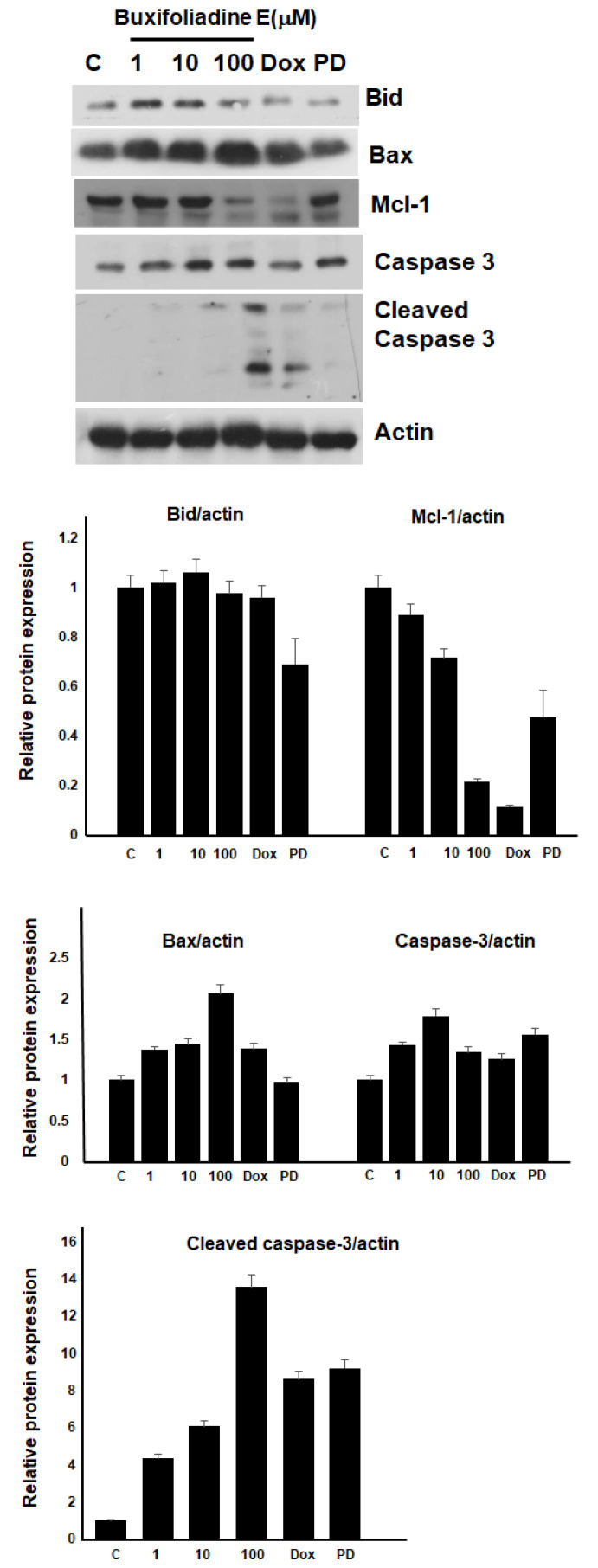
Buxifoliadine E induces apoptosis of HepG2 cells.

**Figure 9 molecules-27-03865-f009:**
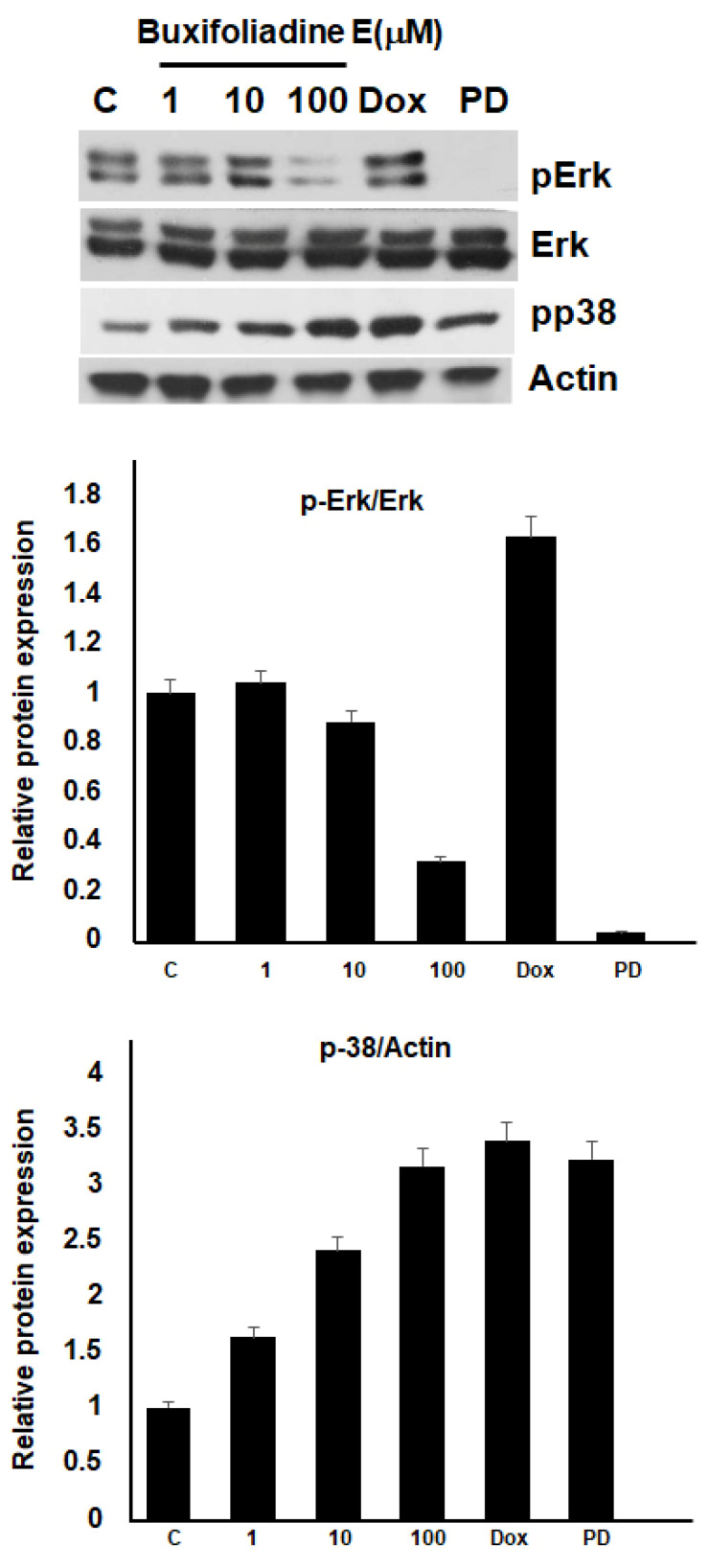
Buxifoliadine E induces apoptosis of HepG2 cells through the Erk pathway.

**Figure 10 molecules-27-03865-f010:**
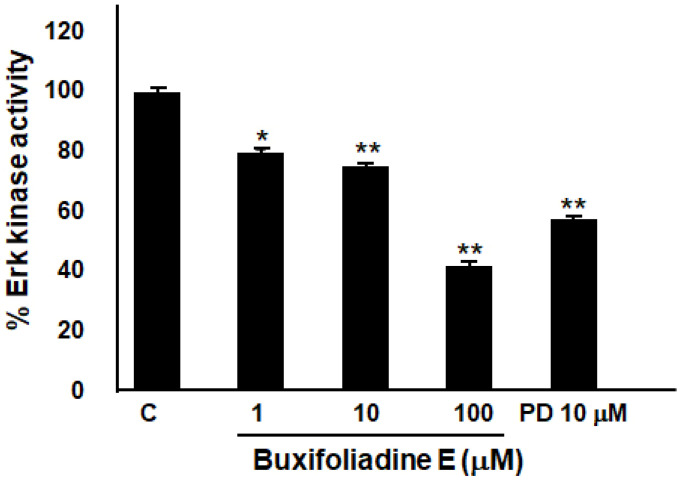
Effect of buxifoliadine E on Erk enzyme-kinase activity. * *p* < 0.05, ** *p* < 0.01.

**Figure 11 molecules-27-03865-f011:**
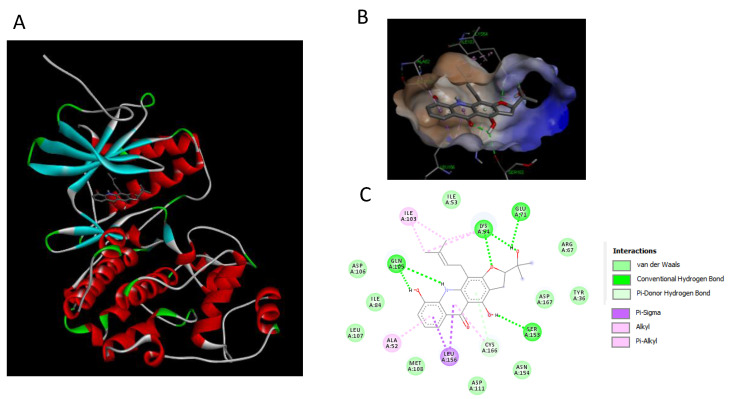
Binding modes and binding-interaction diagrams of buxifoliadine E bound to Erk. (**A**): buxifoliadine E bound at the ATP-binding site of Erk; (**B**): binding interaction between buxifoliadine E and Erk; (**C**): binding-interaction diagrams of buxifoliadine E and Erk.

**Figure 12 molecules-27-03865-f012:**
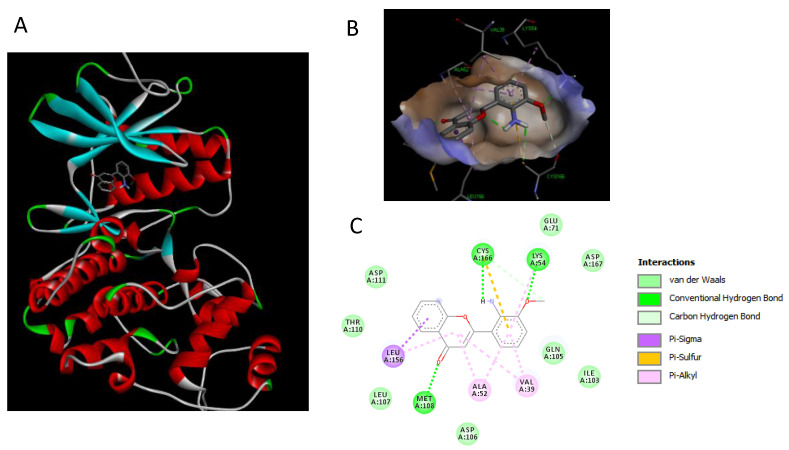
Binding modes and binding-interaction diagrams of Erk inhibitor PD98059, PD bound to Erk. (**A**): PD bound at the ATP-binding site of Erk; (**B**): binding interaction between PD and Erk; (**C**): binding-interaction diagrams of PD and Erk.

**Figure 13 molecules-27-03865-f013:**
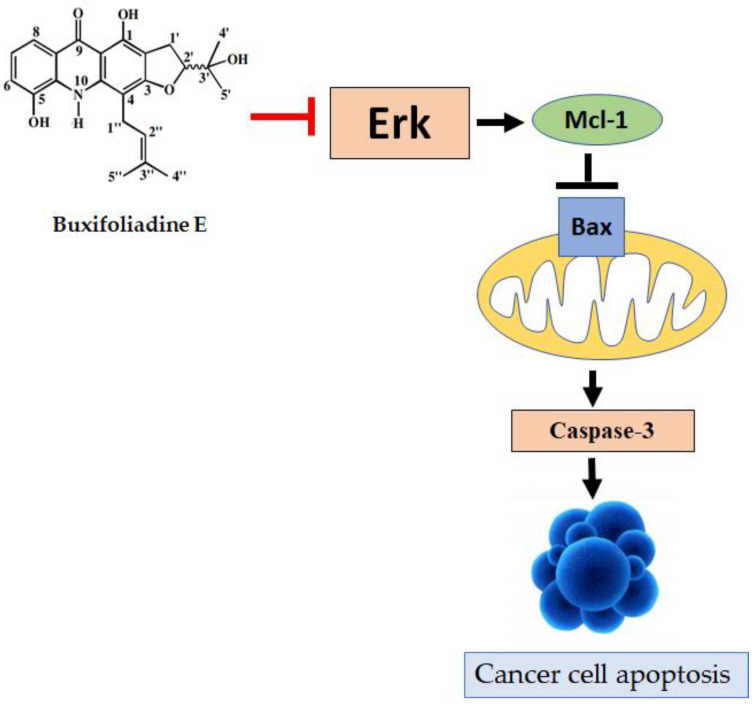
Buxifoliadine E inhibits cell proliferation and induces apoptosis through Erk pathways.

**Table 1 molecules-27-03865-t001:** Pharmacokinetic profile of buxifoliadine E predicted by in silico analysis (pKCSM program).

Pharmacokinetic Properties	Model Name	Predicted Value
Absorption	Human intestinal bsorption (% absorbed)	68.876
Distribution	VDss (human) (log L/kg) (L/kg)	0.255 (1.8 L/kg)
Metabolism	CYP2D6 substrate	No
	CYP3A4 substrate	No
	CYP1A2 inhibitor	No
	CYP2C19 inhibitor	Yes
	CYP2C9 inhibitor	No
	CYP2D6 inhibitor	No
	CYP3A4 inhibitor	No
Excretion	Total clearance (log mL/min/kg)	0.198
	Renal OCT2 substrate	No
Toxicity	Hepatotoxicity	No

## Data Availability

The data presented in this study are available on request from the corresponding authors.

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
