# Peer review of "Acridone Derivatives from *Atalantia monophyla* Inhibited Cancer Cell Proliferation through ERK Pathway"

_molecules, 2022, doi:10.3390/molecules27123865_

Round 1

Reviewer 1 Report

The comments raised were in general properly addressed. I have, however, to ask authors to pay attention into one aspect. The new subsection added in the discussion section expects to discuss the data obtained and not merely to repeat that included in the results section. Have other authors already assessed the ADMET of this compound? There have been similarities or discrepancies in such findings? Also, and comparing to other promising compounds, in this one safer or not? 

Author Response

Dear Reviewer 1,

Thank you for your comments the manuscript entitled “Acridone Derivatives from Atalantia monophyla Inhibited Cancer Cell Proliferation through ERK Pathway” submitted online for publication in Molecules.

The manuscript has been revised as follows.

The comments raised were in general properly addressed. I have, however, to ask authors to pay attention into one aspect. The new subsection added in the discussion section expects to discuss the data obtained and not merely to repeat that included in the results section. Have other authors already assessed the ADMET of this compound? There have been similarities or discrepancies in such findings? Also, and comparing to other promising compounds, in this one safer or not? 

               -  There are few publications about buxifoliadine E and we discussed from references no. 15 and 24

   -The in silico investigation of ADMET revealed that human intestinal absorption of acridones ranged from 68.876% to 99.555. (Takomthong P. et. al., 2021)

Yours sincerely,

Pornthip Waiwut, Ph.D.

Division of Biopharmacy,

Faculty of Pharmaceutical Sciences,

Ubon Ratchathani University,  

Ubon Ratchathani, Thailand.

Reviewer 2 Report

Thank you for the correction you have done. The manuscript is improved a lot. Before accepting the manuscript for publication in Molecules, Apoptosis/necrosis ratio of buxifoliadine E should be calculated at IC50 concentration on the most sensitive cells HepG2. Check figure 3 of this article published in molecules. (Molecules 2018, 23, 59; doi:10.3390/molecules23010059)

Author Response

Dear Reviewer 2,

Thank you for your comments the manuscript entitled “Acridone Derivatives from Atalantia monophyla Inhibited Cancer Cell Proliferation through ERK Pathway” submitted online for publication in Molecules.

The manuscript has been revised as follows.

Thank you for the correction you have done. The manuscript is improved a lot. Before accepting the manuscript for publication in Molecules, Apoptosis/necrosis ratio of buxifoliadine E should be calculated at IC50 concentration on the most sensitive cells HepG2. Check figure 3 of this article published in molecules. (Molecules 2018, 23, 59; doi:10.3390/molecules23010059)

We calculated the IC50 of buxifoliadine E on cancer cells. Buxifoliadine E at 100 µM significantly induced LNCaP, HepG2, HT29, and SHSY5Y cancer cell death with IC50 43.10, 41.36, 64.60 and 96.27 µM respectively.

Yours sincerely,

Pornthip Waiwut, Ph.D.

Division of Biopharmacy,

Faculty of Pharmaceutical Sciences,

Ubon Ratchathani University,  

Ubon Ratchathani, Thailand.

Reviewer 3 Report

Manuscript ''Acridone Derivatives from Atalantia monophyla Inhibited Cancer Cell Proliferation through ERK Pathway'' reported by Waiwut is suitable for publication after minor revision by considering the following points.

  1. In figure 2, the Y-axis of naming missing the bracket.
  2.  Figure 3 needs to make uniform time and in between space.
  3. Need to mention the concentration of doxorubicin.
  4. Need to expand the conclusion part. 

Author Response

Dear Reviewer 3,

Thank you for your comments the manuscript entitled “Acridone Derivatives from Atalantia monophyla Inhibited Cancer Cell Proliferation through ERK Pathway” submitted online for publication in Molecules.

The manuscript has been revised as follows.

Manuscript ''Acridone Derivatives from Atalantia monophyla Inhibited Cancer Cell Proliferation through ERK Pathway'' reported by Waiwut is suitable for publication after minor revision by considering the following points.

  1. In figure 2, the Y-axis of naming missing the bracket.

We have revised the Figure 2 as following;

  1.  Figure 3 needs to make uniform time and in between space.

We have revised the Figure 3 as following;

  1. Need to mention the concentration of doxorubicin.

The concentration of doxorubicin is mentioned in the manuscript.

  1. Need to expand the conclusion part. 

We have revised discussion as following;

Buxifoliadine E, an acridone alkaloid from Atalantia monophyla act as Erk inhibitor, inhibited Erk enzyme activity, which causes of inhibition of Mcl-1 expression, increasing of Bax which induced caspase-3 activation and apoptosis of cancer cell (Figure 12). Thus, these results suggested that buxifoliadine E is a potential candidate for the development of treatments for cancer. However, further study is still needed to provide more data for the investigation of the specific signaling pathway.

Yours sincerely,

Pornthip Waiwut, Ph.D.

Division of Biopharmacy,

Faculty of Pharmaceutical Sciences,

Ubon Ratchathani University,  

Ubon Ratchathani, Thailand.

Round 2

Reviewer 2 Report

I know that you have buxifoliadine E IC50 against different cell lines but measurement of Apoptosis/necrosis effect of buxifoliadine E should be calculated at IC50 concentration on HepG2 by annexin V staining. It is important to understand its anti proliferative effect. 

Author Response

Dear Reviewer 2,

May 30, 2022

Thank you for your comments the manuscript entitled “Acridone Derivatives from Atalantia monophyla Inhibited Cancer Cell Proliferation through ERK Pathway” submitted online for publication in Molecules.

Response to the comment of Reviewer 2

I know that you have buxifoliadine E IC50 against different cell lines but measurement of Apoptosis/necrosis effect of buxifoliadine E should be calculated at IC50 concentration on HepG2 by annexin V staining. It is important to understand its anti proliferative effect.

Although we appreciate the comment asking to calculate IC50 of Annexin V staining, we do not agree with the scientific value of this additional data over to IC50 value of cell viability that we already provided in the last revision. Considering the nature of Annexin V staining that detecting early apoptotic cells by binding phosphatidylserine, it has been well-known that some, but not all, Annexin V positive cells fell into terminal cell death. Therefore, we strongly believe calculating IC50 value of Annexin V assay won’t provide any scientific significance over to IC50 of cell viability data to understand their anti-proliferative effects of the presented data. If the reviewer think providing IC50 data of Annexin V assay is truly necessary to support our conclusion of this study, please explain more in detail and convince us the scientific merit of this data.

Yours sincerely,

Pornthip Waiwut, Ph.D.

Division of Biopharmacy,

Faculty of Pharmaceutical Sciences,

Ubon Ratchathani University,  

Ubon Ratchathani, Thailand.
